# NeuroMamba: A Universal Spatiotemporal Module for Robust Perception in Degraded Sensory Streams

**Jinfeng Li** [1]  **Huijia Song** [1]  **Xiangyue Hu** [1]  **HanLiang Zhou** [2]  **Jiahui Zhang** [1]  **Xinpeng Jiang** [1]  **Fangli Guan** [1]  **Bin Lin** [1]  **Dingran Dong** [1]  **Liqi Yan** [1]  **Pan Li** [1 3]

## Abstract

In open-world intelligent systems, processing continuous sensory streams disrupted by heterogeneous degradation sources presents a fundamental challenge: reconciling the inherent tension between observational completeness and reconstruction fidelity. Methods that prioritize completeness by bridging long-term occlusions often introduce spurious artifacts, while approaches that focus on aggressive noise suppression inevitably disrupt temporal continuity and erase valid structures. To address this challenge, we propose NeuroMamba, a universal plug-and-play module that enhances spatiotemporal consistency in degraded streams. NeuroMamba tackles the dual objectives through two synergistic components. First, we introduce a regional Hybrid Spatiotemporal Rectification (HSR) module, which leverages Mamba-based inertial modeling of linear complexity to recover short-horizon temporal dependencies and infer missing modalities under partial observability. Second, we design a Spiking Confidence Gate (SCG) that enforces reconstruction fidelity under occupancy-guided supervision. Implemented as a hard-thresholding spiking gate unit based on leaky integrate-and-fire (LIF) neurons, SCG distinguishes valid geometric features from sensor noise based on accumulated temporal evidence. Extensive experiments on the nuScenes robustness benchmark demonstrate that NeuroMamba effectively reconciles the trade-off between completeness and fidelity, outperforming the performance of existing approaches in restoring high-fidelity spatiotemporal features from severely incomplete and degraded observations.

---

[1]School of Computer Science, Hangzhou Dianzi University, Hangzhou, China. [2]The Peddie School, Hightstown, NJ, USA. [3]Zhejiang Institute of Artificial Intelligence, Hangzhou, China. Correspondence to: Pan Li <lipan@ieee.org>.

*Proceedings of the 43rd International Conference on Machine Learning*, Seoul, South Korea. PMLR 306, 2026. Copyright 2026 by the author(s).

## 1. Introduction

In open-world perception tasks, maintaining spatiotemporal consistency from continuous sensor streams is fundamental for robust decision-making in autonomous systems like robots, surveillance networks, and self-driving vehicles (Liu et al., 2024a). However, real-world data streams are often disrupted by non-stationarity, sensor failures, transmission delays, and environmental occlusions (Dong et al., 2023; Zang et al., 2019; Tang et al., 2025). Such disturbances shatter the continuity of the feature manifold, rendering downstream tasks vulnerable.

With the continuous advance in multimodal sensors (e.g., LiDAR, vision, and audio arrays), autonomous systems now possess rich perceptual capabilities (Sezgin et al., 2023). Yet, in practical applications, these sensors frequently encounter data loss or degeneration. For example, LiDAR point clouds may exhibit distance-dependent sparsity. Due to hardware malfunction, certain sensor data or video frames could be missing, and parts of visual images could be occluded (Wei et al., 2025). Traditional fusion mechanisms, which assume perfect data alignment and temporal consistency, struggle under such conditions (Mao et al., 2023). When data become degraded, these mechanisms lack the predictive capacity to infer missing modalities from partial observations, making real-time processing of lossy data a persistent challenge. Particularly, in multimodal fusion, the ability to seamlessly impute missing parts while maintaining system efficiency and accuracy has emerged as a critical algorithmic imperative (Zhang et al., 2025; Korse et al., 2024).

Existing restoration approaches face a fundamental trade-off between achieving observational completeness and maintaining reconstruction fidelity. Addressing observational incompleteness requires retrieving sufficient historical context to bridge persistent occlusions. While discriminative sequence models such as Transformers offer precise context modeling (Selva et al., 2023), the quadratic computational complexity of $O(L^2)$ restricts their applicability to short temporal windows (Khan et al., 2022). This computational bottleneck renders them ineffective for recovering from extended sensor failures where critical retrieval cues may lie

beyond the accessible receptive field (Wen et al., 2022). Simultaneously, managing reconstruction uncertainty demands distinguishing valid geometric features from sensor noise. While generative models such as diffusion networks can denoise data, they introduce stochastic uncertainties and substantial inference latency (Ho et al., 2020). Moreover, their potential to synthesize plausible but non-existent artifacts presents severe risks, especially in safety-critical environments (Jabbour & Janapa Reddi, 2024). Similarly, deterministic extrapolation methods lack semantic selectivity, indiscriminately propagating environmental artifacts and sensor noise into the fused representation (Ke et al., 2025). Consequently, there is a lack of a unified module capable of achieving both efficient data retrieval and robust, noise-suppressed reconstruction.

To bridge this gap, we propose NeuroMamba, a plug-and-play module that enforces spatiotemporal consistency across degraded sensory streams. NeuroMamba reconciles the aforementioned dichotomy through a synergistic two-stage pipeline. First, to tackle observational incompleteness, we develop a Hybrid Spatiotemporal Rectification (HSR) module that exploits the linear complexity of Mamba (Gu & Dao, 2024) to retrieve motion inertia from short-horizon history for imputing missing dynamics. This is augmented by intra-frame cross-attention to preserve local geometric continuity (Vaswani et al., 2017; Mei et al., 2020), thereby preventing spatial tearing in rectified features. Second, to mitigate reconstruction uncertainty, we design a Spiking Confidence Gate (SCG). Unlike generative models prone to probabilistic artifacts, an SCG functions as a neuromorphic filter. By utilizing leaky integrate-and-fire dynamics (Wu et al., 2018), it selectively propagates only high-fidelity features to effectively suppress transient sensor noise and hallucinations. Finally, the rectified features are fused via a residual connection (He et al., 2016), ensuring robust perception without disrupting the backbone's inference flow. Our contributions are summarized as follows.

- We propose NeuroMamba, a plug-and-play module designed to address feature incompleteness via spatiotemporal imputation. By leveraging historical context to compensate for degraded signals, it seamlessly integrates into Bird's-Eye-View (BEV) pipelines to recover robust perception with competitive inference efficiency.

- We devise two synergistic components, i.e., HSR and SCG. The HSR module addresses observational incompleteness by employing Mamba-driven inertial extrapolation to impute missing features from historical context. Complementing this, an SCG serves as a neuromorphic filter leveraging integrate-and-fire dynamics to selectively propagate geometrically consistent features and strictly bound reconstruction uncertainty.

- We construct a robustness benchmark based on nuScenes by synthesizing degraded evaluation sets across varying spatial sparsity and continuous occlusion (Caesar et al., 2020). Extensive experiments on this benchmark validate that NeuroMamba sets a new state-of-the-art, significantly outperforming baselines in reconstruction fidelity under degradation scenarios while maintaining competitive inference overhead.

**Conflict of Interest Disclosure.** The authors declare no financial conflicts of interest. This work was conducted in an academic setting and is not affiliated with any commercial entity whose products are evaluated in this paper.

## 2. Related Work

### 2.1. Temporal Inertia vs. Computational Complexity

While Transformers dominate the perception domain (Vaswani et al., 2017), a fundamental bottleneck stems from their global self-attention mechanism where computing pairwise token interactions defined by the $QK^T$ matrix incurs strict quadratic complexity. To preserve real-time capability, systems are compelled to truncate historical context, thereby severing temporal cues essential for resolving persistent occlusions. Sub-quadratic alternatives such as Linear Transformers and early State Space Models mitigate this latency but introduce distinct mechanical flaws (Wang et al., 2020; Peng et al., 2023; Neumeyer et al., 2010; Fu et al., 2023). Linear variants approximate the attention map via kernel-based feature maps that often degrade retrieval precision, while traditional State Space Models operate as Linear Time-Invariant systems governed by static transition matrices. These input-independent dynamics reduce the model to a passive filter that indiscriminately integrates sensor noise alongside valid features. While Mamba addresses this static limitation through an input-dependent selective scan (Gu & Dao, 2024), it operates as a continuous-valued system reliant on multiplicative soft gating. This mechanism merely dampens low-amplitude sensor noise rather than eliminating it, inevitably allowing residual artifacts to leak into the hidden state (Zhu et al., 2024; Liu et al., 2024b; Li et al., 2024). Our approach overcomes this deficiency by integrating neuromorphic spiking dynamics to enforce a physical hard threshold that strictly suppresses the transient artifacts standard Mamba cannot filter.

### 2.2. Robust Perception and Feature Restoration

Feature restoration aims to recover the underlying physical state from degraded observations, crucial for maintaining state continuity under sensor failure in autonomous systems (Xie et al., 2018). Current approaches to feature restoration can be classified into generative and discrim-

inative categories, yet both confront fundamental structural limitations in safety-critical domains (Zhang et al., 2022; Shen et al., 2025). Generative frameworks such as Diffusion Probabilistic Models rely on an iterative reverse sampling mechanism to reconstruct data distributions from Gaussian noise (Wei et al., 2023; Wang et al., 2024a; Hu et al., 2023). This dependence on probabilistic denoising introduces intrinsic indeterminacy that risks synthesizing high-fidelity artifacts without physical counterparts while the heavy computational cost of multi-step inference renders these methods incompatible with real-time decision loops (Aithal et al., 2024; Chen et al., 2022). Conversely, discriminative approaches optimize for deterministic texture completion but typically process spatial tokens as independent entities to conserve computation (Chen et al., 2024; Plizzari et al., 2021; Zhou et al., 2023). The absence of global manifold constraints in this isolation strategy leads to geometric misalignment where independently extrapolated regions fail to maintain structural coherence across the field of view. Beyond restoration modules, recent work has also studied robust perception architectures under sensor failure. MetaBEV (Ge et al., 2023) handles complete modality dropout via Switched Modality Training. MoME (Park et al., 2025) routes queries to modality-specific expert decoders. UniBEV (Wang et al., 2024b) unifies multi-modal fusion with built-in robustness to missing sensors. These methods mainly operate at the modality-routing level and target individual modality failure. In contrast, NeuroMamba operates *post-fusion*, recovering spatiotemporal consistency from historical frames even when all modalities degrade simultaneously.

## 2.3. Spiking Neural Networks

Information filtering in modern sequence models is primarily governed by analog gating mechanisms to manage feature persistence. Models such as Gated Recurrent Units (GRU) and Gated Linear Units (GLU) utilize continuous activation functions like Sigmoid or Tanh to modulate the flow of features (Dey & Salem, 2017; Hochreiter & Schmidhuber, 1997). These gates learn to attenuate irrelevant information by mapping inputs to a continuous range allowing for end-to-end optimization of the filtering strategy. The fundamental flaw of this approach lies in the mathematical continuity of the gating functions which remain non-zero almost everywhere. Spiking Neural Networks (SNNs) offer a paradigm shift: unlike Artificial Neural Networks (ANNs) that process continuous floating-point values (Wang, 2003), SNNs operate on discrete binary spikes, thereby enabling sparse and potentially energy-efficient computation (Wu et al., 2018; Gerstner et al., 2014). However, although surrogate gradient learning has enabled deep SNNs training, most studies mainly focus on energy efficiency on neuromorphic hardware, treating SNNs merely as low-power approxima-

tions of standard networks. Rather than focusing strictly on energy efficiency, our work leverages the discrete mathematical properties of SNN dynamics as an algorithmic utility to construct a hard-threshold feature gate.

## 3. Methodology

We propose NeuroMamba, a neuromorphic state space module for robust perception under data degeneration. As illustrated in Figure 1, NeuroMamba functions as a plug-and-play module preceding the detection backbone and consists of two synergistic components: (1) a Hybrid Spatiotemporal Rectification (HSR) component that leverages Mamba's selective inertia to extrapolate missing dynamics, and (2) a Spiking Confidence Gate (SCG) component that utilizes leaky integrate-and-fire (LIF) dynamics to filter reconstruction noise.

### 3.1. Problem Formulation

We model the sequence of degraded sensory streams as $\mathcal{X} = \{\mathbf{x}_t\}_{t=1}^T$, where $\mathbf{x}_t \in \mathbb{R}^{H \times W \times C}$ denotes the degraded feature map at time step $t$, subject to heterogeneous degradation. The primary objective is to learn a universal mapping function $\mathcal{F}_\theta$ that reconstructs the underlying spatiotemporally consistent manifold $\hat{\mathbf{x}}_t$ from these incomplete observations. This restoration process is modeled as:

$$\hat{\mathbf{x}}_t = \mathcal{F}_\theta(\mathbf{x}_t, \mathbf{h}_{t-1}), \qquad (1)$$

where $\mathbf{h}_{t-1}$ denotes the latent state capturing historical motion priors, and $\mathcal{F}_\theta$ is the proposed NeuroMamba module parameterized by $\theta$.

### 3.2. Hybrid Spatiotemporal Rectification (HSR)

The HSR aims to mitigate observational sparsity by incorporating a strong inductive bias of motion inertia. To reconcile the inherent trade-off between computational efficiency and local geometric fidelity, we adopt a divide-and-conquer strategy, decomposing the restoration process into a cascaded four-stage transformation: (1) latent space projection for compressing noisy, sparse observations into a compact, robust representation, (2) inertial dynamics modeling for extrapolating missing object dynamics and repairing degraded features, (3) spatial consistency restoration for re-establishing global geometric coherence and smoothing out boundary artifacts, and (4) hierarchical manifold decoding for reconstructing high-fidelity feature patches.

**Latent Space Projection via Hierarchical Encoding.** To decouple structural semantics from high-frequency stochastic noise, the input $\mathbf{x}_t$ is spatially partitioned into local patches $\mathbf{x}_t^{(i)}$, which are then projected onto a compact latent space $\mathcal{Z}$ via a query-based encoder:

$$\mathbf{z}_t^{(i)} = \text{CrossAttn}(\mathbf{Q}_{learn}, \text{StridedConv}(\mathbf{x}_t^{(i)})), \quad (2)$$

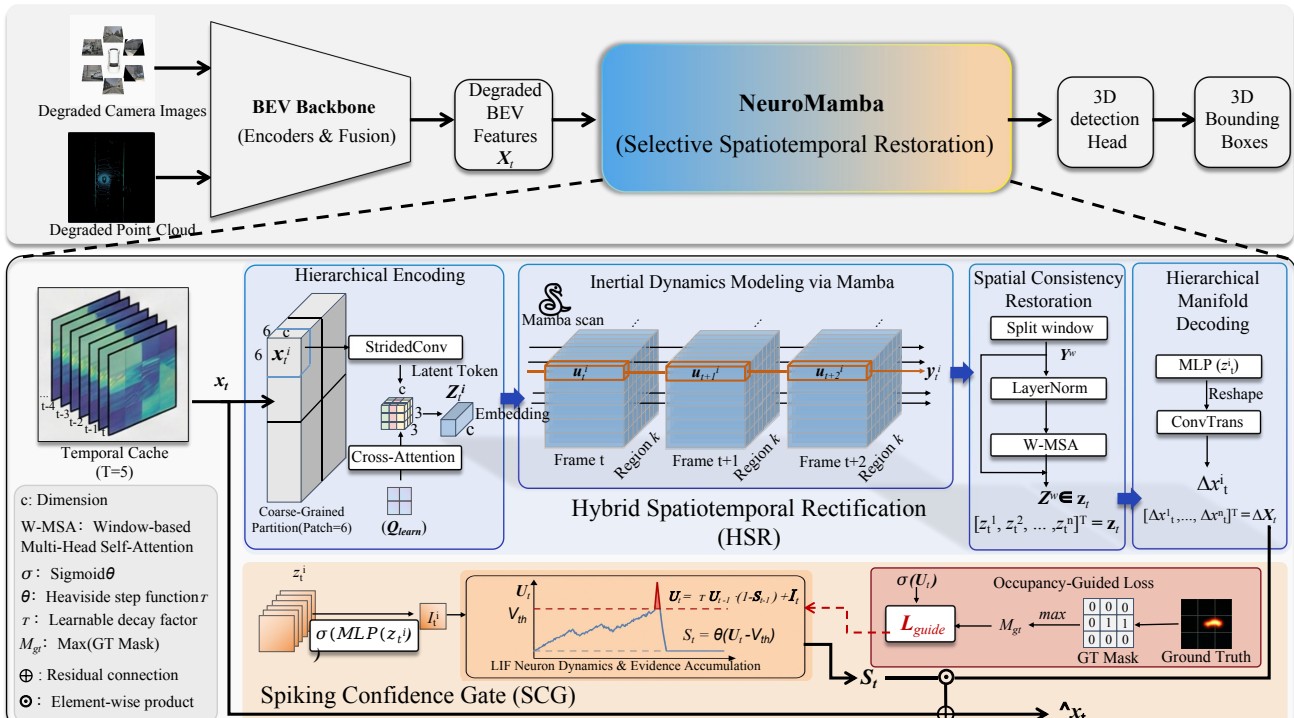

*Figure 1.* An overview of the NeuroMamba architecture. It consists of two synergistic components: a Hybrid Spatiotemporal Rectification (HSR) module for inertial feature extrapolation and a Spiking Confidence Gate (SCG) module for neuromorphic noise suppression.

where $\mathbf{Q}_{learn} \in \mathbb{R}^{N \times D}$ denotes a set of learnable latent queries designed to aggregate semantic context, and StridedConv($\cdot$) functions as a local feature extractor to capture neighborhood geometry. This operation compresses the sparse observation into a dense latent token $\mathbf{z}_t^{(i)} \in \mathbb{R}^D$. effectively establishing the robust initial state for the subsequent dynamic modeling.

**Inertial Dynamics Modeling via Selective SSM.** To efficiently capture motion dynamics within a $T$-frame window while minimizing parameter redundancy, we propose a regional shared inertial modeling mechanism. A critical challenge in adopting shared weights is the potential loss of positional context. To resolve this ambiguity, we first inject an explicit 3-level hierarchical embedding into the latent tokens, establishing a strict spatiotemporal correspondence:

$$\mathbf{u}_t^{(i)} = \mathbf{z}_t^{(i)} + \mathbf{E}_{\text{time}}^{(t)} + \mathbf{E}_{\text{spatial}}^{(i)} + \mathbf{E}_{\text{region}}^{(k)}, \quad (3)$$

where $\mathbf{u}_t^{(i)}$ is the position-aware token, $\mathbf{E}_{\text{time}}^{(t)}$, $\mathbf{E}_{\text{spatial}}^{(i)}$, and $\mathbf{E}_{\text{region}}^{(k)}$ are the frame, patch, and region embeddings, respectively, with $i \in \mathcal{R}_k$. Subsequently, the collected tokens $\{\mathbf{u}_t^{(i)}\}$ from all regions are treated as parallel sequences and scanned along the temporal axis by a single Mamba block. This design allows the model to learn a universal motion prior that is invariant to spatial location yet sensitive to local geometry.

Assuming object motion follows physical inertia, we model the latent evolution as a continuous-time system discretized via the zero-order hold (ZOH) principle. We employ the selective State Space Model (SSM) to approximate this evolution independently for each location $i$:

$$\begin{aligned}\mathbf{h}_t^{(i)} &= \overline{\mathbf{A}}_t(\mathbf{u}_t^{(i)})\mathbf{h}_{t-1}^{(i)} + \overline{\mathbf{B}}_t(\mathbf{u}_t^{(i)})\mathbf{u}_t^{(i)}, \\ \mathbf{y}_t^{(i)} &= \mathbf{C}\mathbf{h}_t^{(i)} + \mathbf{D}\mathbf{u}_t^{(i)}.\end{aligned} \quad (4)$$

Here, $\mathbf{h}_t^{(i)}$ and $\mathbf{y}_t^{(i)}$ are the inertial state and output token of patch $i$, $\bar{\mathbf{A}}_t(\mathbf{u}_t^{(i)})$ and $\bar{\mathbf{B}}_t(\mathbf{u}_t^{(i)})$ are input-dependent discretized SSM matrices, and $\mathbf{C}$ and $\mathbf{D}$ are readout and skip projections. This formulation acts as an implicit confidence gate: when the input exhibits degradation patterns (e.g., occlusion), the selective mechanism drives $\overline{\mathbf{B}}_t \to 0$ while maintaining $\overline{\mathbf{A}}_t \approx \mathbf{I}$, where $\mathbf{I}$ denotes the identity matrix. By decoupling the state update from unreliable observations, the system evolves strictly from the historical state $\mathbf{h}_{t-1}^{(i)}$. This enables closed-loop trajectory extrapolation that maintains physical momentum and continuity even when sensor data is missing.

**Spatial Consistency Restoration.** While the inertial dynamics modeling (Eq. 4) ensures temporal continuity, the region-wise independence of the SSMs implies that the updated features $\mathbf{y}_t^{(i)}$ may lack global spatial coherence. To enforce geometric consistency, we first reassemble the lo-

cal tokens into a global feature map $\mathbf{y}_t \in \mathbb{R}^{N \times D}$. We partition $\mathbf{y}_t$ into a set of non-overlapping local windows $\{\mathbf{Y}^{(1)}, \ldots, \mathbf{Y}^{(K)}\}$, where each $\mathbf{Y}^{(w)} \in \mathbb{R}^{M^2 \times D}$ represents the features within the $w$-th window. We then apply Window-based Multi-Head Self-Attention (W-MSA) (Liu et al., 2021) to model local geometric affinity strictly within each window:

$$\mathbf{Z}^{(w)} = \text{W-MSA}\big(\text{LN}(\mathbf{Y}^{(w)})\big) + \mathbf{Y}^{(w)}, \qquad (5)$$

where $\text{LN}(\cdot)$ denotes Layer Normalization. This residual formulation preserves the inertial priors carried by $\mathbf{Y}^{(w)}$ while the attention mechanism learns a spatial correction term to smooth out boundary artifacts. Finally, the refined windows $\mathbf{Z}^{(w)}$ are merged back to the original grid to form the globally consistent latent map $\mathbf{z}_t$. Formally, this reconstruction establishes the correspondence between the local window refinements and the global token sequence:

$$\mathbf{z}_t = \mathcal{P}^{-1}\left(\{\mathbf{Z}^{(w)}\}_{w=1}^K\right) = \left[\mathbf{z}_t^{(1)}, \ldots, \mathbf{z}_t^{(N)}\right]^\top, \quad (6)$$

where $\mathcal{P}^{-1}$ denotes the inverse window partitioning operator, and $\mathbf{z}_t^{(i)}$ represents the refined feature vector for the $i$-th spatial location. By restricting interactions to local windows (e.g., $M = 5$), we reduce the computational complexity from $\mathcal{O}(N^2)$ to $\mathcal{O}(N \cdot M^2)$, ensuring efficiency on high-resolution feature grids.

**Hierarchical Manifold Decoding.** To reconstruct the feature patches from the latent space $\mathcal{Z}$, we design a hierarchical patch decoder. Direct linear projection from a 1D token to a 2D patch is ill-posed and often prone to checkerboard artifacts. To mitigate this, we adopt a coarse-to-fine upsampling strategy. Specifically, the latent token $\mathbf{z}_t^{(i)}$ is first projected to a coarse structural grid (e.g., $3 \times 3$) via a multilayer perceptron (MLP). Subsequently, a transposed convolution is applied to upsample this grid to the target resolution (e.g., $6 \times 6$) with refined details. The decoding process for the residual correction term $\Delta\mathbf{x}_t^{(i)}$ is formulated as:

$$\Delta\mathbf{x}_t^{(i)} = s \cdot \text{ConvTrans}\Big(\text{Reshape}(\mathbf{MLP}(\mathbf{z}_t^{(i)}))\Big), \quad (7)$$

where $s$ is a learnable scale, $\text{MLP}$ is a multilayer perceptron, $\text{Reshape}$ maps vectors to coarse grids, and $\text{ConvTrans}$ denotes transposed convolution.

### 3.3. Spiking Confidence Gate (SCG)

While the HSR provides an inertial estimate of the latent dynamics, indiscriminate fusion of these extrapolated features with raw observations risks introducing reconstruction artifacts or hallucinating non-existent objects. To mitigate this, we introduce the Spiking Confidence Gate (SCG), a neuromorphic filtering mechanism that selectively integrates restored features based on accumulated temporal evidence.

**Synaptic Current Transduction.** Unlike standard SNNs that process raw sensory signals, our SCG operates on the refined latent manifold. We first define a synaptic transduction layer to map the high-dimensional latent token $\mathbf{z}_t^{(i)}$ (from Eq. 6) into a scalar input current $I_t^{(i)}$. This is implemented via a lightweight MLP with a function $\sigma(\cdot)$:

$$I_t^{(i)} = \sigma(\text{MLP}(\mathbf{z}_t^{(i)})), \qquad (8)$$

where $I_t^{(i)} \in (0, 1)$ quantifies the instantaneous potential validity of the inferred feature at location $i$, derived from its spatiotemporal context.

**Temporal Evidence Accumulation via LIF.** We repurpose the Leaky Integrate-and-Fire (LIF) neuron model (Gerstner et al., 2014) as a temporal evidence accumulator. The membrane potential $\mathbf{U}_t^{(i)}$ integrates the synaptic current over time, governed by the discrete difference equation:

$$\mathbf{U}_t^{(i)} = \tau \mathbf{U}_{t-1}^{(i)} \cdot (1 - \mathbf{S}_{t-1}^{(i)}) + I_t^{(i)}, \qquad (9)$$

where $\mathbf{U}_t^{(i)}$ is the membrane potential acting as a temporal-evidence accumulator, $\tau \in [0, 1]$ is a learnable decay factor, and $\mathbf{S}_{t-1}^{(i)}$ is the previous spike output that resets the accumulated potential upon firing. This dynamic formulation confers unique temporal selectivity to the module. Specifically, for high-fidelity signals, the rapid accumulation of $I_t$ allows the potential to promptly surpass the firing threshold $V_{th}$, ensuring an immediate response to valid entities. Conversely, for ambiguous or partially occluded features, the neuron requires evidence integration over multiple frames to trigger a spike, thereby providing a delayed confirmation that effectively filters out transient noise. For background regions dominated by stochastic artifacts, the leakage term $\tau$ ensures that sub-threshold potentials continuously dissipate, preventing the erroneous opening of the gate due to sporadic activations.

**Spike-Gated Residual Integration.** A binary spike $\mathbf{S}_t^{(i)}$ is generated via the Heaviside step function:

$$\mathbf{S}_t^{(i)} = \Theta(\mathbf{U}_t^{(i)} - V_{th}), \qquad (10)$$

where $V_{th}$ denotes the firing threshold, and $\Theta(\cdot)$ is the Heaviside step function which outputs 1 if the argument is non-negative and 0 otherwise. This binary spike serves as a localized confidence mask. Finally, we explicitly reassemble the local decoded residuals $\Delta\mathbf{x}_t^{(i)}$ (derived in Eq. 7) into a global residual map $\Delta\mathbf{X}_t$. We align the low-resolution global spike map $\mathbf{S}_t$ (composed of all $\mathbf{S}_t^{(i)}$) with the high-resolution feature space via nearest-neighbor upsampling. The final restoration is performed by injecting the gated residuals into the global raw observation $\mathbf{x}_t$:

$$\hat{\mathbf{x}}_t = \mathbf{x}_t + \mathbf{S}_t \odot \Delta\mathbf{X}_t. \qquad (11)$$

where $\odot$ is the element-wise multiplication operator. By this design, the SCG strictly decouples valid geometric restoration from stochastic noise, ensuring that the final reconstruction $\hat{\mathbf{x}}_t$ preserves the fidelity of raw observations while selectively filling in missing dynamics with high-confidence inertial estimates.

### 3.4. Optimization and Occupancy-Guided Supervision

**Patch-wise Occupancy Ground-Truth Construction.** To strictly align supervision with the tokenized patch grid, we propose a coarse-grained alignment strategy. 3D ground-truth annotations are projected onto the BEV plane to form a binary occupancy map, which is then downsampled via max-pooling to the patch resolution, yielding the mask $\mathbf{M}_{gt} \in \{0,1\}^N$. A patch is labeled active (1) if it encompasses any object-occupied pixels, ensuring strict spatial synchronization with the SNN's latent manifold.

**Membrane Potential Guided Activation Loss.** Directly calculating loss on the sparse, discrete spikes $\mathbf{S}_t \in \{0,1\}$ can lead to gradient vanishing or instability. To enable effective gradient flow, we shift the supervision target from the discrete output to the continuous membrane potential $\mathbf{U}_t$. The membrane potential serves as a proxy for the neuron's accumulated confidence regarding object existence. We map the unbounded potential $\mathbf{U}_t^{(i)}$ to the $(0,1)$ probability interval via a Sigmoid function and compute the Weighted Binary Cross-Entropy (WBCE) loss against the ground-truth mask $\mathbf{M}_{gt}$:

$$\mathcal{L}_{guide} = \frac{1}{N} \sum_{i=1}^{N} \text{WBCE}\Big(\sigma(\mathbf{U}_t^{(i)}), \mathbf{M}_{gt}^{(i)}\Big), \quad (12)$$

The WBCE term incorporates a balancing weight to mitigate background dominance. Physically, this objective imposes a dual constraint by compelling neurons in object regions to accumulate high potentials ($\mathbf{U}_t \to +\infty$) for deterministic firing while simultaneously suppressing background potentials ($\mathbf{U}_t \to -\infty$) to inhibit false alarms. This continuous supervision enforces geometrically consistent patterns and effectively circumvents the non-differentiability of the Heaviside step function.

**End-to-End Total Objective.** The entire NeuroMamba module is trained end-to-end in conjunction with the downstream task. The total objective is formulated as:

$$\mathcal{L}_{total} = \mathcal{L}_{task} + \lambda_{guide}\mathcal{L}_{guide}, \quad (13)$$

where $\mathcal{L}_{task}$ represents standard task-specific objectives modulated by the coefficient $\lambda_{guide}$. Guided by $\mathcal{L}_{guide}$, the SCG module rapidly converges to distinguish foreground regions in early training stages and provides precise gating signals for feature fusion. This auxiliary term effectively acts as a regularizer that aligns the SNN's firing manifold with the physical geometric occupancy of the scene.

## 4. Experiments

In this section, we evaluate NeuroMamba on the task of multi-modal 3D object detection, a representative case study for restoring degraded spatiotemporal feature streams.

### 4.1. Benchmark: nuScenes-D Construction

Our evaluation is based on nuScenes (Caesar et al., 2020), which we extend into nuScenes-D to test spatiotemporal resilience beyond generic noise. Our benchmark is independent from the nuScenes-C benchmark of Dong et al. (Dong et al., 2023), which applies per-frame i.i.d. corruptions. In contrast, nuScenes-D employs physically grounded degradation and a temporally coherent burst protocol. Detailed configurations are summarized in Table 1. This benchmark features a hierarchical degradation pipeline and a non-i.i.d. temporal corruption protocol, ensuring simulated failures reflect real-world physical and temporal dynamics.

**Hierarchical Degradation Pipeline.** Diverging from stochastic 2D perturbations, we formulate a two-stage pipeline that emulates the physical formation of sensory data. In the first stage, we simulate volumetric environmental effects based on atmospheric optics, synthesizing depth-dependent fog governed by Koschmieder's Law (Koschmieder, 1925) and precipitation-induced streaks via Mie scattering dynamics (Mie, 1908). Building upon this environmental layer, the second stage implements hardware-specific failures, including 11 types of camera imaging artifacts, such as localized occlusions and dynamic distortions, alongside 7 categories of LiDAR structural information loss, ranging from sector dropout to beam scattering.

Crucially, to move beyond naive stochastic noise, we implement a Temporal Burst Protocol with a window size of $\Delta t = 3$. By freezing the layout seeds within each window, this protocol ensures that corruptions are non-i.i.d. and temporally coherent, thereby faithfully emulating persistent physical disturbances like mechanical jitter or lens blockage.

*Table 1.* Quantitative Parameters and Statistical Coverage of nuScenes-D.

| Degradation Level | Light | Medium | Heavy | Extreme |
|---|---|---|---|---|
| Target Coverage (Config) | 15.0% | 35.0% | 60.0% | 85.0% |
| Avg. Sensor Coverage (Actual) | 14.1% | 29.7% | 58.7% | 81.7% |
| Max Concurrent Effects | 1 | 2 | 3 | 4 |
| Fog Extinction Coeff. $\beta$ | 0.04 | 0.08 | 0.12 | 0.18 |
| Rain Streak Count $N$ | 1200 | 2400 | 4000 | 6500 |
| **Dataset Corruption Rate** | **70.30%** | **90.35%** | **97.77%** | **100.00%** |

### 4.2. Evaluation Metrics and Experimental setup

**Evaluation Metrics.** We follow the official nuScenes evaluation protocol using mean Average Precision (mAP) and the

nuScenes Detection Score (NDS) as primary metrics. To facilitate a comprehensive assessment of restoration quality regarding localization and kinematic estimation, we further report auxiliary metrics including mean Average Translation (mATE), Scale (mASE), Orientation (mAOE), Velocity (mAVE), and Attribute (mAAE) errors.

**Experimental setup.** Experiments were performed on 6 NVIDIA RTX 4090D GPUs over 32 epochs. NeuroMamba processes BEV features at a $180 \times 180$ resolution with 256 channels. For the HSR, we employ a single Mamba layer with a hidden state dimension of 16.

### 4.3. Comparison with State-of-the-Art Methods

We benchmark against two methods: BEVFusion (Liu et al., 2023) (as the baseline backbone to verify the capacity to empower static pipelines with dynamic rectification for degraded streams) and BEVDiffuser (Ye et al., 2025) (a state-of-the-art diffusion-based plug-and-play module to evaluate the comparative strength of our restoration capabilities). We further benchmark against dedicated robust perception systems, including UniBEV (Wang et al., 2024b) and MoME (Park et al., 2025), to evaluate NeuroMamba's performance in robustness-oriented perception scenarios.

**Performance under Degradation.** As summarized in Table 2 and Figure 2, equipping BEVFusion with NeuroMamba yields strong resilience across all degradation levels. While vanilla BEVFusion collapses as degradation intensifies, with mAP dropping from 61.6 to 7.8 under *Extreme*, the NeuroMamba-augmented model retains a clear margin throughout. In the *Medium* setting, it gains +48.3% mAP and +22.3% NDS over the baseline, and consistently surpasses the BEVDiffuser plug-in.

**Evaluation on Clean Dataset.** On the original *Clean* dataset, NeuroMamba remains competitive as a plug-in. As Table 2 shows, applied to BEVFusion it reaches the highest mAP of 62.7, ahead of vanilla BEVFusion at 61.6 and the BEVDiffuser plug-in at 62.5. Its NDS of 58.2 drops only marginally from the baseline value of 59.7, showing that robustness is gained without sacrificing clean-data accuracy.

**Inference Efficiency.** We compare the computational overhead of NeuroMamba and BEVDiffuser, both operating as plug-in modules on the same BEVFusion backbone. As shown in Table 3, NeuroMamba achieves around $33\times$ fewer FLOPs and $10\times$ lower latency with significantly fewer parameters, confirming that the performance gains reported in Table 2 do not come at a prohibitive computational cost.

**Comparison with Robust Perception Systems.** We further compare against dedicated robust perception systems under medium degradation. Unlike NeuroMamba, which operates as a post-fusion plug-in, MoME (Park et al., 2025) and UniBEV (Wang et al., 2024b) are end-to-end architectures

with modality-level routing. As shown in Table 4, even as a lightweight plug-in on BEVFusion, NeuroMamba achieves stronger performance on the main detection metrics under medium degradation, with higher mAP and NDS as well as lower mATE and mAVE.

### 4.4. Ablation Study

In our ablation study, we focus on the specific contribution of each component, as well as the impact of feature scales and the spatiotemporal granularity within different temporal windows under medium-level degradation.

**Component Contribution.** Table 5 isolates the impact of the HSR and SCG modules. The baseline BEVFusion struggles to maintain perceptual integrity under data degradation, yielding an NDS of only 43.1 and an mAP of 35.4. The sole integration of the Mamba-based HSR boosts the metrics to 46.1 NDS and 41.3 mAP, validating its capability to extrapolate missing temporal dynamics and restore observational completeness. Building on this, the inclusion of SCG further elevates the performance to 52.7 NDS and 52.5 mAP. These results suggest that while pure temporal extrapolation may introduce noise, the neuromorphic filtering provided by SCG effectively suppresses it and maintains the fidelity of the reconstructed features.

**Gating Mechanism Comparison.** To examine the role of the neuromorphic gate, we replace SCG with four ANN gates of comparable parameter count ($\sim$60–65K), namely Sigmoid, CNN, GRU, and Transformer, under the same HSR backbone and training setup. As shown in Table 6, Sigmoid performs worst and even underperforms the no-gate variant, indicating that a simple soft gate is not effective under degraded observations. GRU and Transformer improve the results, but they still fall well below SCG in mAP and NDS. This suggests that the gain mainly comes from membrane-potential evidence accumulation and binary firing, rather than parameter count alone.

**Firing Threshold Sensitivity.** We evaluate the sensitivity of the LIF firing threshold under medium degradation. As shown in Table 7, $V_{\text{th}} = 1.0$ gives the best result. Lower thresholds admit weakly supported candidates, while higher thresholds suppress both noise and valid recoveries, leading to worse performance. Although the threshold is fixed, the response remains input-dependent because different degradation levels change the HSR output quality and thus the input current.

**Feature Scale Selection.** We investigate the impact of spatial granularity on feature restoration by partitioning BEV feature maps into varying patch scales of $2\times 2$, $4\times 4$, and $6\times 6$. While we initially hypothesized that a multi-scale fusion strategy (Scales 2, 4, 6) would enhance multi-granularity perception, Table 8 reveals that the single-scale configura-

*Table 2.* Comparison of detection performance on nuScenes-val under sensor degradation. Values in parentheses denote the relative percentage improvement over the baseline BEVFusion (Liu et al., 2023).

| Condition | mAP (↑) | | | NDS (↑) | | |
|---|---|---|---|---|---|---|
| | BEVFusion(Liu et al., 2023) | + BEVDiffuser(Ye et al., 2025) | +NeuroMamba | BEVFusion(Liu et al., 2023) | + BEVDiffuser(Ye et al., 2025) | +NeuroMamba |
| Clean | 61.6 | 62.5 | **62.7** (+1.80%) | 59.7 | **60.2** | 58.2 (-2.50%) |
| Light | 49.7 | 49.9 | **55.5** (+11.7%) | 49.5 | 52.0 | **53.8** (+8.70%) |
| Medium | 35.4 | 38.1 | **52.5** (+48.3%) | 43.1 | 44.7 | **52.7** (+22.3%) |
| Heavy | 22.8 | 22.4 | **30.7** (+34.6%) | 34.3 | 34.1 | **39.9** (+16.3%) |
| Extreme | 7.8 | 5.8 | **9.9** (+26.9%) | 17.3 | 17.6 | **27.0** (+56.1%) |

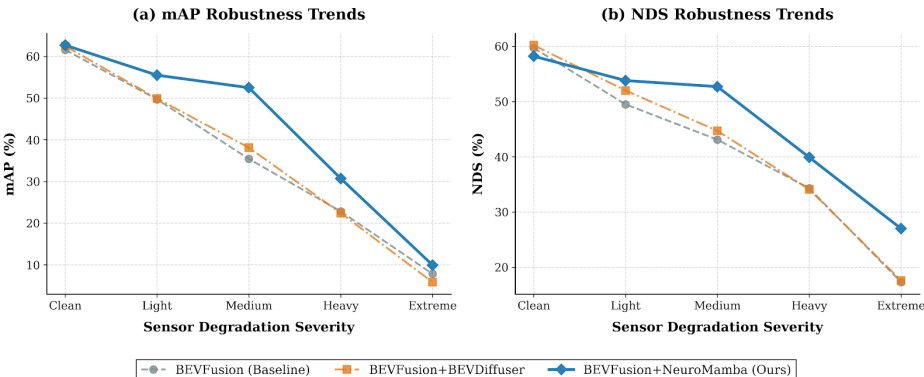

*Figure 2.* Robustness trends under different sensor degradation levels. (a) mAP performance trends of different methods; (b) NDS performance trends of different methods.

*Table 3.* Inference efficiency comparison between plug-in modules on the BEVFusion backbone.

| Module | Params(M) | FLOPs(G) | Latency(ms) | Mem(GB) |
|---|---|---|---|---|
| BEVDiffuser | 59.96 | 1413.2 | 84.19 | **0.345** |
| Ours | **7.995** | **42.8** | **8.27** | 0.661 |

*Table 4.* Comparison with dedicated robust perception systems.

| Method | Scores (↑) | | TP Errors (↓) | | | | |
|---|---|---|---|---|---|---|---|
| | mAP | NDS | mATE | mASE | mAOE | mAVE | mAAE |
| MoME (Park et al., 2025) | 49.3 | 52.1 | 0.382 | **0.259** | 0.368 | 1.127 | **0.242** |
| UniBEV (Wang et al., 2024b) | 46.6 | 51.0 | 0.375 | 0.265 | **0.346** | 1.120 | 0.250 |
| Ours | **52.5** | **52.7** | **0.334** | 0.262 | 0.510 | **1.047** | 0.247 |

tion of Scale 6 achieves superior performance with 52.5 mAP and 52.7 NDS. In contrast, Scale 4 exhibits the most severe decay (47.5 mAP, 50.1 NDS), while the multi-scale ensemble yields a sub-optimal 50.2 mAP. These results underscore a critical trade-off between reconstruction precision and noise introduction: although finer or multi-scale partitions aim to capture diverse object dimensions, they inadvertently introduce semantic interference and spatial misalignment that outweigh the benefits of multi-scale representation. Consequently, we adopt Scale 6 to maximize semantic robustness while minimizing observational noise.

**Temporal Modeling Analysis.** We replace the Mamba block inside HSR with a temporal Transformer of matched depth and comparable parameter count (∼8.0M). As shown in Table 9, from $T=5$ to $T=20$, NeuroMamba's FLOPs increase by $3.4\times$ (near-linear) versus $4.8\times$ for the Transformer (super-linear). While the Transformer achieves lower latency at $T=5$ due to optimized GPU attention kernels, this reverses at $T=20$ (28.86 ms vs. 22.69 ms). The Mamba-based HSR also consistently achieves higher mAP and NDS at every window size, confirming that the efficiency gain does not sacrifice perception quality. Performance peaks at $T=5$ for both variants, as extended history exacerbates cumulative ego-pose misalignment that outweighs the benefit of additional context.

### 4.5. Visualization Analysis

We visualize feature-level rectification in Figure 3.

**Precision of SCG Activation.** Comparing the spiking activations in panel (b) with the ground truth (c), the patches generated by SCG are far from stochastic noise. Instead, they align precisely with object-dense zones. This high spatial consistency validates the module's capability to accurately localize regions of high semantic value.

**Rectification of Missed Detections.** Comparing panel (b) against the baseline in (a) visually highlights NeuroMamba's impact. Our module successfully recovers targets missed by the baseline, with these detections aligning precisely with the module's patching range. This confirms that NeuroMamba effectively compensates for feature degradation, recalling critical targets that the baseline failed to capture.

*Table 5.* Component ablation analysis. Values in parentheses denote relative improvements over the baseline BEVFusion.

| Methods | Scores (↑) | | TP Errors (↓) | | | | |
|---|---|---|---|---|---|---|---|
| | mAP | NDS | mATE | mASE | mAOE | mAVE | mAAE |
| BEVFusion (Liu et al., 2023) | 35.4 | 43.1 | 0.342 | 0.266 | 0.543 | 1.184 | 0.307 |
| BEVFusion + HSR | 41.3 (+16.7%) | 46.1 (+7.0%) | 0.339 (-0.9%) | **0.262** (-1.5%) | 0.535 (-1.5%) | 1.232 (+4.1%) | 0.313 (+2.0%) |
| BEVFusion + HSR&SCG | **52.5** (+48.3%) | **52.7** (+22.3%) | **0.334** (-2.3%) | 0.262 (-1.5%) | **0.510** (-6.1%) | **1.047** (-11.6%) | **0.247** (-19.5%) |

*Table 6.* Comparison of gating mechanisms under medium degradation.

| Gate | Scores (↑) | | TP Errors (↓) | | | | |
|---|---|---|---|---|---|---|---|
| | mAP | NDS | mATE | mASE | mAOE | mAVE | mAAE |
| No gate | 41.3 | 46.1 | 0.339 | 0.262 | 0.535 | 1.232 | 0.313 |
| Sigmoid (MLP) | 36.5 | 41.5 | 0.385 | 0.262 | 0.550 | 1.250 | 0.285 |
| CNN | 40.5 | 46.1 | 0.355 | 0.262 | 0.520 | 1.180 | 0.275 |
| GRU | 43.3 | 48.0 | 0.338 | 0.262 | 0.498 | 1.085 | 0.264 |
| Transformer | 44.2 | 48.9 | **0.326** | 0.262 | **0.478** | 1.055 | 0.258 |
| SCG | **52.5** | **52.7** | 0.334 | **0.262** | 0.510 | **1.047** | **0.247** |

*Table 7.* Impact of the firing threshold $V_{th}$ on detection performance.

| $V_{th}$ | 0.5 | 1.0 (Ours) | 1.5 | 1.75 | 2.0 |
|---|---|---|---|---|---|
| mAP (↑) | 49.5 | **52.5** | 48.2 | 50.2 | 46.0 |
| NDS (↑) | 51.1 | **52.7** | 49.4 | 51.1 | 47.6 |
| mAVE (↓) | 1.085 | **1.047** | 1.140 | 1.095 | 1.180 |

*Table 8.* Impact of input scale configurations. Quantitative comparison of detection performance across different scale settings.

| Scales | Scores (↑) | | TP Errors (↓) | | | | |
|---|---|---|---|---|---|---|---|
| | mAP | NDS | mATE | mASE | mAOE | mAVE | mAAE |
| 2 | 51.8 | 52.2 | **0.333** | **0.262** | 0.510 | 1.114 | 0.266 |
| 4 | 47.5 | 50.1 | 0.335 | 0.263 | **0.495** | 1.053 | 0.267 |
| 6 | **52.5** | **52.7** | 0.334 | **0.262** | 0.510 | **1.047** | **0.247** |
| 2, 4, 6 | 50.2 | 50.8 | 0.342 | 0.264 | 0.553 | 1.124 | 0.274 |

*Table 9.* Mamba- and Transformer-based HSR comparison across temporal window sizes $T$.

| Module | $T$ | FLOPs (G) | Latency (ms) | FPS | Mem (GB) | Scores (↑) | |
|---|---|---|---|---|---|---|---|
| | | | | | | mAP | NDS |
| Ours (Mamba) | 5 | 42.8 | 8.27 | 121.0 | 0.661 | **52.5** | **52.7** |
| | 10 | 77.7 | 13.41 | 74.6 | 1.182 | **51.0** | **51.4** |
| | 20 | 147.5 | 22.69 | 44.1 | 2.219 | **47.0** | **49.2** |
| Transformer | 5 | 48.2 | 6.46 | 154.9 | 0.664 | 50.7 | 51.1 |
| | 10 | 99.2 | 13.50 | 74.1 | 1.181 | 49.2 | 50.0 |
| | 20 | 233.6 | 28.86 | 34.7 | 2.219 | 46.8 | 48.8 |

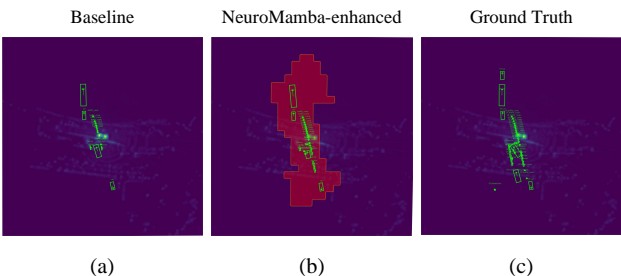

| Baseline | NeuroMamba-enhanced | Ground Truth |
|---|---|---|
| (a) | (b) | (c) |

*Figure 3.* Visualization of feature-level detection results. Panel (a) shows the baseline BEVFusion output, (b) represents the results enhanced by NeuroMamba, and (c) provides the Ground Truth.

## 5. Limitations

NeuroMamba has clear operating boundaries. First, HSR relies on motion inertia from recent frames, so inertial candidates may lag when objects undergo sudden maneuvers. SCG mitigates this by requiring multi-frame evidence, but cannot guarantee correctness in all cases. Second, persistent structured noise could accumulate sufficient membrane potential to trigger false spikes. The occupancy-guided loss (Eq. 12) makes this unlikely but cannot be entirely ruled out. Third, performance degrades under extreme corruption, and NeuroMamba should be deployed within a redundant perception stack in safety-critical systems.

## 6. Conclusion

We present NeuroMamba, a plug-and-play module designed to ensure robust spatiotemporal perception under severe sensor degradation. By synergizing the long-horizon efficiency of Mamba-based inertial modeling with the strict noise-suppression capabilities of neuromorphic spiking dynamics, NeuroMamba effectively rectifies heavily corrupted sensory streams while preserving high reconstruction fidelity, significantly outperforming existing robust perception frameworks. Future research will focus on developing advanced ego-pose alignment strategies to unlock even longer effective temporal windows, as well as integrating probabilistic generative priors to handle scenarios of near-total information loss.

## Acknowledgments

This work was supported by the Zhejiang Provincial Natural Science Foundation of China [grant number LQN25D010005] and the National Natural Science Foundation of China [grant number 42401517].

## Impact Statement

This work advances robust perception in degraded sensory streams, with applications in autonomous driving and mobile robotics. By restoring spatiotemporal consistency under sensor failures and adverse conditions, NeuroMamba contributes to safer operation in scenarios where conventional pipelines fail. As a safety-critical component, it should be deployed within a redundant perception stack rather than as a sole safeguard, and real-world deployment warrants validation beyond the nuScenes benchmark.

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
