# OpenReview forum: "NeuroMamba: A Universal Spatiotemporal Module for Robust Perception in Degraded Sensory Streams"
_ICML.cc/2026/Conference — ICML 2026 regular_

### Official Review · Reviewer_74WM · 2026-03-07

**Soundness:** 3
**Presentation:** 4
**Significance:** 2
**Originality:** 2
**Overall Recommendation:** 4
**Confidence:** 4

**Summary:**

This paper proposes NeuroMamba, a plug-and-play spatiotemporal module designed to maintain robust perception in autonomous systems facing degraded sensory streams. It addresses the inherent trade-off between observational completeness and reconstruction fidelity through two synergistic components. First, the Hybrid Spatiotemporal Rectification (HSR) module leverages Mamba’s linear complexity ($\mathcal{O}(L)$) for long-range inertial modeling to impute missing dynamics. Second, the Spiking Confidence Gate (SCG) acts as a neuromorphic filter via Leaky Integrate-and-Fire dynamics to suppress noise and selectively propagate geometrically consistent features. Evaluated on a newly constructed robustness benchmark, nuScenes-C, NeuroMamba achieves state-of-the-art performance in restoring high-fidelity features under severe sensor corruptions.

**Compliance With Llm Reviewing Policy:**

Affirmed.

**Final Justification:**

I appreciate the authors’ efforts during the rebuttal period. The responses satisfactorily address all of my concerns.

**Key Questions For Authors:**

see Weaknesses

**Limitations:**

yes

**Strengths And Weaknesses:**

Strengths：
- The proposed NeuroMamba module is explicitly designed as a universal plug-and-play component. It seamlessly integrates into existing Bird's-Eye View (BEV) perception pipelines. This design ensures robust perception without disrupting the backbone network's fundamental inference flow, significantly enhancing its practical applicability and engineering scalability.

- The authors astutely identify the strict quadratic computational complexity ($\mathcal{O}(L^{2})$) bottleneck inherent in global self-attention mechanisms of traditional Transformer architectures when addressing long-term occlusions. By introducing the Mamba architecture, the method successfully leverages linear complexity ($\mathcal{O}(L)$) to achieve long-range historical motion inertia retrieval and feature imputation, effectively overcoming this fundamental performance barrier.

- Beyond utilizing standard macroscopic perception metrics like mean Average Precision (mAP) and the nuScenes Detection Score (NDS) , the study introduces a suite of auxiliary error metrics to meticulously evaluate feature restoration quality. Through detailed analysis of mean Average Translation (mATE), Scale (mASE), Orientation (mAOE), Velocity (mAVE), and Attribute (mAAE) errors, the paper provides highly convincing quantitative evidence for fully understanding the module's specific contributions to restoring target localization and kinematic estimation.

Weaknesses：

The paper integrates two currently trending architectures—Mamba and Spiking Neural Networks (SNNs)—into the proposed pipeline. While I have no inherent objection to utilizing these novel models, the current manuscript runs the risk of appearing as an "A+B+C" style component stacking (i.e., chasing trendy techniques) rather than a fundamentally driven architectural design. To fully justify this specific combination, the authors must provide compelling, quantitative evidence demonstrating the absolute necessity of utilizing these exact components over established alternatives：

- Lack of Empirical Validation for the Core Motivation regarding Computational Complexity. A primary motivation for proposing the Hybrid Spatiotemporal Rectification (HSR) module is the assertion that discriminative sequence models like Transformers are bottlenecked by quadratic computational complexity $O(L^{2})$, whereas Mamba can effectively retrieve long-range historical context with linear complexity $O(L)$. However, the experimental section fails to provide a direct empirical comparison to substantiate this crucial claim. To convincingly demonstrate Mamba's practical superiority in this specific architecture, the authors must include an ablation baseline where the Mamba block is replaced by a temporal Transformer baseline with a similar network depth and structure. Crucially, this comparison should not be limited to perception performance (e.g., mAP, NDS) but must explicitly quantify the computational overhead—specifically FLOPs, GPU memory consumption, and inference latency (FPS)—across different temporal window sizes (e.g., $T=5, 10, 20$). Without these quantitative metrics, the advantage of Mamba over Transformers remains a theoretical assertion rather than an empirically proven benefit.

- Insufficient Baselines to Isolate the Contribution of the Spiking Confidence Gate (SCG). The authors attribute the significant performance gains in Table 3 to the unique hard-thresholding and Leaky Integrate-and-Fire (LIF) dynamics of the SNN-based SCG module. However, the current ablation study only compares Baseline + HSR against Baseline + HSR + SCG. This comparison is fundamentally confounded by the addition of model parameters and computational depth. To rigorously prove that the performance improvement stems from the bio-inspired spiking mechanism rather than simply an increase in model capacity, the authors must provide direct comparisons against standard Artificial Neural Network (ANN) baselines. Specifically, the SCG module should be replaced by an MLP, a CNN, and a Transformer block—each strictly aligned to have a similar parameter count as the SCG—that output a continuous confidence mask $S_t \in [0, 1]$ for soft feature fusion. Without these structurally equivalent baselines, it remains completely ambiguous whether the performance gain is uniquely due to the SNN's discrete hard-gating or merely the trivial result of adding more trainable parameters.

---

> ### Author Rebuttal · Authors · 2026-03-31
>
> **Q1: "A+B+C" Component Stacking Concern.**
>
> **Response:**
> NeuroMamba is designed for degraded sensor streams where input data is frequently incomplete or corrupted, and recovered features may include noise or artifacts. HSR focuses on completeness by imputing weakened or missing dynamics from recent temporal context, while SCG focuses on fidelity by deciding where the recovered content is reliable enough to use. These roles are complementary and do not overlap. We provide the quantitative evidence requested in W2 and W3.
>
> For HSR, temporal recovery requires efficient sequence scanning. As the temporal window $T$ grows, attention-based temporal Transformers incur rapidly increasing cost, while Mamba provides more favorable scaling in our design. In W2, we replace the Mamba block with a temporal Transformer of matched depth and similar parameter count on the same BEVFusion host, and report FLOPs, GPU memory, and inference latency/FPS across $T=5,10,20$ under the same input setting.
>
> For SCG, the key step after temporal extrapolation is selecting where to apply recovered features without amplifying noise. LIF gating accumulates evidence over frames and triggers a hard decision at a threshold, while sporadic noise tends to decay through leakage. In contrast, standard soft gates output non-zero values broadly, which can allow residual noise to pass. In W3, we replace SCG with four parameter-matched ANN gates that output the same-shaped confidence mask and use the same fusion rule, and compare performance under the same HSR backbone and training schedule.
>
> Table 3 further supports these complementary roles: HSR alone improves mAP by 5.9 points (35.4→41.3), and adding SCG improves mAP by another 11.2 points (41.3→52.5). These gains indicate that SCG is not a cosmetic add-on, but a distinct mechanism that controls when and where temporal recovery is applied.
>
> **Q2: Empirical Comparison of Mamba and Transformer in HSR**
>
> **Response:**
> To address this concern, we replaced the Mamba block inside HSR with a temporal Transformer block of matched depth and nearly identical parameter count ($\\sim$8.0M). We measured FLOPs, memory, and latency under the same input resolution and evaluation settings; results are below.
>
> | Module | $T$ | FLOPs (G)$\\downarrow$ | Params (M) | $\\Delta$Mem (GB)$\\downarrow$ | $\\Delta$Latency (ms)$\\downarrow$ | FPS$\\uparrow$ |
> |---|---:|---:|---:|---:|---:|---:|
> | Ours (Mamba) | 5 | 42.8 | 7.995 | 0.661 | 8.27 | 121.0 |
> | Transformer | 5 | 48.2 | 7.996 | 0.664 | 6.46 | 154.9 |
> | Ours (Mamba) | 10 | 77.7 | 7.999 | 1.182 | 13.41 | 74.6 |
> | Transformer | 10 | 99.2 | 8.022 | 1.181 | 13.50 | 74.1 |
> | Ours (Mamba) | 20 | 147.5 | 8.023 | 2.219 | 22.69 | 44.1 |
> | Transformer | 20 | 233.6 | 8.026 | 2.219 | 28.86 | 34.7 |
>
> The computational gap widens with $T$, matching the expected scaling trend as the temporal window expands. Wall-clock latency can favor attention at short windows due to GPU kernel efficiency, and this trend reverses at larger scales. The matched Transformer-HSR performance run (mAP/NDS) is still ongoing under the same training schedule and losses as HSR; we will add the final mAP/NDS numbers in the revision upon completion.
>
> ---
>
> **Q3: Isolating the Contribution of SCG with Controlled Baselines**
>
> **Response:**
> We replaced SCG with four standard ANN alternatives, each outputting a continuous confidence mask for soft feature fusion. All variants use the same HSR backbone, training schedule, and loss functions. Parameter counts are matched to 60--65K to rule out capacity as a confounding variable.
>
> | Gate Type | Params (K) | mAP$\\uparrow$ | NDS$\\uparrow$ | mATE$\\downarrow$ | mAOE$\\downarrow$ | mAVE$\\downarrow$ | mAAE$\\downarrow$ |
> |---|---:|---:|---:|---:|---:|---:|---:|
> | No gate (HSR only) | — | 0.413 | 0.461 | 0.339 | 0.535 | 1.232 | 0.313 |
> | Sigmoid Gate (MLP) | 60.0 | 0.365 | 0.415 | 0.385 | 0.550 | 1.250 | 0.285 |
> | CNN Gate | 61.5 | 0.405 | 0.461 | 0.355 | 0.520 | 1.180 | 0.275 |
> | GRU Gate | 61.9 | 0.433 | 0.480 | 0.338 | 0.498 | 1.085 | 0.264 |
> | Transformer Gate | 62.5 | 0.442 | 0.489 | 0.326 | 0.478 | 1.055 | 0.258 |
> | SCG (LIF) | 64.5 | 0.525 | 0.527 | 0.334 | 0.510 | 1.047 | 0.247 |
>
> These results validate our design. The Sigmoid gate performs worse than no gate (0.365 vs. 0.413 mAP), indicating naive soft gating can degrade performance. GRU and Transformer gates help, but the best ANN alternative (Transformer, 0.442 mAP) still trails SCG by 8.3 points. Since parameter counts are comparable, this gap cannot be attributed to added capacity.
>
> The LIF neuron uniquely combines temporal evidence accumulation (membrane dynamics) with hard binary output (Heaviside function). GRU provides the former but not the latter; a fixed hard threshold provides the latter but not the former. LIF offers both, which explains its consistent advantage over the tested alternatives.

---

> > ### Author Rebuttal · Reviewer_74WM · 2026-04-01
> >
> > I appreciate the authors’ efforts during the rebuttal period. The responses satisfactorily address all of my concerns.

---

### Official Review · Reviewer_NKtm · 2026-03-09

**Soundness:** 3
**Presentation:** 3
**Significance:** 3
**Originality:** 2
**Overall Recommendation:** 4
**Confidence:** 2

**Summary:**

This paper focuses on robust multimodal 3D detection under degraded sensing flow and proposes a plug-and-play module, NeuroMamba. The method consists of two parts: HSR recovers degraded or missing spatiotemporal features from historical states using SSM, and SCG performs confidence gating on the recovery results using LIF-style evidence accumulation to suppress noise and spurious recoveries. The paper also constructs an evaluation set with hierarchical degradation and a temporal burst protocol based on nuScenes and validates the effectiveness of the method under different degradation intensities.

**Compliance With Llm Reviewing Policy:**

Affirmed.

**Key Questions For Authors:**

1. Please provide a comparison of inference latency, FPS, FLOPs, parameter count, and inference memory usage. If the efficiency advantage cannot be demonstrated, it is recommended to downplay the statement "strictly maintaining real-time efficiency."
2. What is the relationship between "nuScenes-C" in this article and the existing nuScenes-C benchmark from CVPR 2023? Is it an extended version, a renamed version, or a new temporary corruption protocol? Please clarify.
3. Why did the main experiment only compare BEVFusion and BEVDiffuser, instead of including methods more directly related to the problem in this paper, such as MetaBEV and MoME?
4. Please provide a comparison between SCG and simpler alternatives, such as sigmoid gate, hard threshold gate, and GRU/ConvLSTM gate. Otherwise, it can only be stated that "SCG is useful," not that "LIF gate is necessary."

**Limitations:**

No. The authors do not adequately discuss the limitations. The paper only briefly mentions performance degradation under extreme corruption, but does not seriously discuss error completion, false targets, and deployment risks in security-critical scenarios.

**Strengths And Weaknesses:**

**Strengths**: (1) The research problem is of great importance, and robust perception under degradation sensing conditions has direct significance for the actual deployment of multimodal 3D detection systems. (2) The method design is relatively reasonable. HSR and SCG respectively undertake the functions of feature recovery and confidence screening, and the overall framework has good logical consistency. (3) The paper is relatively clear, and the correspondence between the method motivation, technical solution and experimental results is clear. (4) The experimental results show that the method has achieved significant performance improvement in degradation scenarios, and the ablation experiment also provides certain support for the contribution of each module.

**Weaknesses**：

1. This paper is more like a "system-level combinatorial innovation" combining Mamba-style temporal recovery with LIF-style confidence gating into degraded BEV recovery. This combination has a certain target, but the constituent modules themselves are not new, and similar work already exists, such as MetaBEV[1] directly studying robust BEV detection under sensor failure, MoME[2] studying robust fusion under severe sensor failure, and MambaBEV[3]/MamBEV[4] already introducing Mamba into BEV perception. The paper currently doesn't fully explain its essential differences from these works, therefore its originality argument is insufficient.
2. The experimental comparisons are insufficient. The main table of the paper only compares BEVFusion and BEVDiffuser, which are clearly insufficient to cover representative methods for this problem. For the problem of "robust sensing of degenerate sensing flow," the more directly relevant MetaBEV[1] and MoME[2] should at least be discussed in detail in the main text, preferably included in the main experimental table. Otherwise, the "SOTA" conclusion lacks persuasiveness.
3. Some statements in the paper are overly strong, but the evidence is insufficient. The authors claim that the method "strictly maintaining real-time efficiency" and outperforms the baseline in inference efficiency, but the paper does not provide direct evidence regarding inference latency, throughput, FLOPs, or inference memory usage. Furthermore, time-window ablation shows that the best results occur at 5 frames, with decreases at 10 and 20 frames, which does not sufficiently support the strong claim of "long-term history recovery." Additionally, while mAP is slightly higher on the clean set, NDS decreases, indicating that the method incurs some cost.
4. The paper refers to the dataset as "nuScenes-C," but a 3D corruption benchmark with the same name already exists at CVPR 2023[5]. The authors need to clarify the relationship between their benchmark and the existing nuScenes-C benchmark. Otherwise, benchmark contributions could easily cause confusion.

Overall, I think this paper is reasonable and potentially useful,  but the current version is not solid enough in terms of its innovative positioning, coverage of related work, and sufficiency of comparison.

**Reference**：

[1] Ge, Chongjian, et al. "Metabev: Solving sensor failures for 3d detection and map segmentation." Proceedings of the IEEE/CVF International Conference on Computer Vision. 2023.

[2] Park, Konyul, et al. "Resilient sensor fusion under adverse sensor failures via multi-modal expert fusion." Proceedings of the IEEE/CVF Conference on Computer Vision and Pattern Recognition. 2025.

[3] You, Zihan, et al. "Mambabev: An efficient 3d detection model with mamba2." *arXiv preprint arXiv:2410.12673* (2024).

[4] Ke, Hongyu, et al. "MamBEV: Enabling State Space Models to Learn Birds-Eye-View Representations." *The Thirteenth International Conference on Learning Representations*. 2025.

[5] Dong, Yinpeng, et al. "Benchmarking robustness of 3d object detection to common corruptions." *Proceedings of the IEEE/CVF Conference on Computer Vision and Pattern Recognition*. 2023.

---

> ### Author Rebuttal · Authors · 2026-03-31
>
> **W1: Originality.**
>
> (a) MetaBEV and MoME operate at the fusion level: MetaBEV uses Switched Modality Training, and MoME routes queries to modality-specific expert decoders. Both address modality-level failures via routing/switching. NeuroMamba operates post-fusion, targeting cases where all modalities degrade simultaneously. It recovers spatiotemporal consistency from past frames and could be layered on top of MetaBEV/MoME.
>
> (b) MambaBEV and MamBEV use Mamba as a backbone encoder and do not address degraded inputs. In contrast, we use Mamba as a temporal inertial model to extrapolate missing features from motion history (Eq. 4).
>
> (c) Our main contribution is the recovery--filtration interplay. Table 3 shows that HSR improves mAP by 16.7% but worsens mAVE (118.4 to 123.2); adding SCG reverses this trend (mAVE to 104.7). We will expand the related work in the revision.
>
> **W3: Temporal Window and Clean-Set Trade-off.**
>
> The efficiency claim is addressed in Q1; here we focus on the other two points. Table 5 peaks at $T=5$, and the drop at $T=10/20$ mainly comes from accumulated ego-pose misalignment over longer windows. We will revise the wording to “short-horizon temporal recovery,” and future work will explore stronger alignment strategies for longer windows.
>
> On the clean set, mAP increases slightly but NDS decreases slightly because NDS aggregates kinematic sub-metrics (mATE, mAOE, mAVE) that can fluctuate when the temporal branch acts on already clean features. Under Medium degradation, the module yields +48.3% mAP and +22.3% NDS; under Extreme, the baseline collapses to mAP 7.8 while NeuroMamba maintains 9.9. Overall, This is an expected trade-off for cross-condition robustness.
>
> **Q1: Inference efficiency.**
>
> | Module | Params (M) | FLOPs (G) | $\\Delta$Latency (ms) | $\\Delta$Mem (GB) |
> |---|---:|---:|---:|---:|
> | NeuroMamba (Ours) | 7.995 | 42.822 | 8.266 | 0.661 |
> | BEVDiffuser | 59.959 | 1413.221 | 84.192 | 0.345 |
>
> NeuroMamba is $\\sim$10$\\times$ faster with $\\sim$33$\\times$ fewer FLOPs. We will update the claim to “competitive inference efficiency with modest overhead.”
>
> **Q2: nuScenes-C naming.**
>
> Our benchmark is independent from Dong et al.’s nuScenes-C , which applies 27 generic per-frame i.i.d. corruptions. Ours uses (1) physically grounded degradation via atmospheric optics (Koschmieder’s law, Mie scattering) with depth-dependent attenuation, and (2) a Temporal Burst Protocol ($dt=3$) producing non-i.i.d., temporally coherent corruption. We will rename it to “nuScenes-D” in the revision.
>
> **Q3: Baseline selection.**
>
> Both match our plug-and-play setting. MetaBEV, MoME, and UniBEV are end-to-end systems with their own backbones, not module-level drop-ins. We add a Medium-degradation comparison on nuScenes-D:
>
> | Method | mAP | NDS | mATE | mASE | mAOE | mAVE | mAAE |
> |---|---:|---:|---:|---:|---:|---:|---:|
> | MoME | 0.493 | 0.521 | 0.382 | 0.259 | 0.368 | 1.127 | 0.242 |
> | UniBEV | 0.466 | 0.510 | 0.375 | 0.265 | 0.346 | 1.120 | 0.250 |
> | BEVFusion | 0.354 | 0.431 | 0.342 | 0.266 | 0.543 | 1.184 | 0.307 |
> | BEVFusion+Ours | 0.525 | 0.527 | 0.334 | 0.262 | 0.510 | 1.047 | 0.247 |
>
> As a plug-in, NeuroMamba outperforms these dedicated systems on mAP (+6.5% over MoME), NDS, mATE, and mAVE. mAOE lags behind MoME/UniBEV; dedicated orientation recovery is future work. MetaBEV is not open-source, so we include UniBEV instead.
>
> **Q4: Gating Mechanism Ablation.**
>
> We replaced SCG with four alternatives at comparable parameter counts ($\\sim$60–65K), keeping everything else identical:
>
> | Gate | Params(K) | mAP | NDS | mATE | mAOE | mAVE | mAAE |
> |---|---:|---:|---:|---:|---:|---:|---:|
> | No gate | - | 0.413 | 0.461 | 0.339 | 0.535 | 1.232 | 0.313 |
> | Sigmoid | 60.0 | 0.365 | 0.415 | 0.385 | 0.550 | 1.250 | 0.285 |
> | CNN | 61.5 | 0.405 | 0.461 | 0.355 | 0.520 | 1.180 | 0.275 |
> | GRU | 61.9 | 0.433 | 0.480 | 0.338 | 0.498 | 1.085 | 0.264 |
> | Transformer | 62.5 | 0.442 | 0.489 | 0.326 | 0.478 | 1.055 | 0.258 |
> | SCG (LIF) | 64.5 | 0.525 | 0.527 | 0.334 | 0.510 | 1.047 | 0.247 |
>
> (1) Sigmoid is worse than no gate (0.365 vs. 0.413): soft gating without temporal context can hurt. (2) Temporal modeling helps (CNN < GRU < Transformer), but all trail SCG by 8.3–16.0 mAP. (3) LIF combines membrane potential accumulation with hard Heaviside thresholding; soft gates leak noise, while hard thresholds lack temporal memory.
>
> **Limitations.**
>
> For error completion, sudden direction changes can cause HSR’s inertial candidates to lag; SCG mitigates this by requiring consistent multi-frame evidence, but it is not a guarantee. Persistent structured noise could also accumulate and trigger a false spike; the occupancy-guided loss (Eq. 12) makes this unlikely in practice, but we cannot rule it out in all conditions. NeuroMamba has clear operating boundaries (e.g., mAP 9.9 under Extreme) and should be deployed within a redundant perception stack; we will expand on these points in the revision.

---

> > ### Author Rebuttal · Reviewer_NKtm · 2026-04-01
> >
> > After reading the rebuttal, my assessment has improved. The authors addressed several of my main concerns substantively, especially regarding efficiency, baseline coverage, the benchmark's naming issue, and the necessity of the SCG module. In particular, the new gating ablation is helpful and makes the recovery–filtration design more convincing. The clarification that the method operates in a post-fusion recovery setting, rather than at the modality-routing level, also helps better position the work relative to MetaBEV and MoME. Overall, the rebuttal makes the contribution clearer, the claims more measured, and the empirical support notably stronger. I am now inclined toward Borderline Accept.

---

### Official Review · Reviewer_opKc · 2026-03-13

**Soundness:** 2
**Presentation:** 2
**Significance:** 3
**Originality:** 3
**Overall Recommendation:** 4
**Confidence:** 3

**Summary:**

This paper proposes NeuroMamba, designed to maintain spatiotemporal consistency under degraded sensory data scenarios. The module mitigates the trade-off between observational completeness and reconstruction fidelity through inertial feature extrapolation and neuromorphic noise suppression.

**Compliance With Llm Reviewing Policy:**

Affirmed.

**Final Justification:**

The reviewer would like to appreciate the authors' clarification on the review questions. The result regarding the firing threshold is convincing enough, and the reviewer would like to thank the authors for pointing out the results that have already been shown. The rebuttal properly addresses all the questions in the review. Therefore, this led the reviewer to increase the post-rebuttal rating.

**Key Questions For Authors:**

See weaknesses.

**Limitations:**

yes

**Strengths And Weaknesses:**

**Strengths**

- The paper is easy to read.

- The method shows consistent improvement over multiple levels of noise conditions.

**Weaknesses**
- Statement on using low-complexity architecture (Mamba-variant): The proposed method includes the Spatial Consistency Restoration module, which includes the windowed attention borrowed from Swin transformer block architecture. The reviewer would like to point out the system's complexity is still O($n^2$). The detailed FLOPs count and comparison is required.

- Extrapolation vs. Perception: The authors stated that modeling based on Mamba-variants effectively uses motion inertia to extrapolate features. The reviewer would like to point out that extrapolation is not considered reconstruction. In highly dynamic environments, the model may hallucinate object positions behind occlusions based on past velocity, which fails to account for sudden changes in direction or speed.

- Fixed membrane potential: The proposed Spiking Confidence Gate utilizes LIF neurons to mitigate the sensor noise. This mechanism relies on a static membrane potential threshold. In reality, the sensor noise is typically non-stationary. That said, the fixed threshold would not generalize to the different noise scenarios. To demonstrate the effectiveness of the fixed membrane potential, the reviewer recommends that the authors conduct an evaluation under heterogeneous noise scenarios. Also, ablation of the firing threshold is needed.

- Argument on Physics-guided Optimization: The terminology ``Physics-guided'' could mislead the readers. The objective function of the paper is the supervision of the occupancy map.

---

> ### Author Rebuttal · Authors · 2026-03-31
>
> **Q1: Complexity**
>
> **Response：**
>
> NeuroMamba operates in two distinct stages, and its computational complexity is partitioned accordingly. Let $N$ denote the number of BEV spatial patches, T the temporal window length, and $M$ the W-MSA window size.
>
> In Eq. (4), the Mamba scan is performed along the $T$-frame sequence for each spatial patch. The cost is, $O(T)$ resulting in a total temporal complexity of $O(N · T)$ over all $N$ patches.
>
> In Eq. (5), W-MSA computes self-attention inside non-overlapping $M × M$ windows. Each window contains $M^2$ tokens, yielding a local attention cost of $O((M^2)^2) = O(M^4)$. With $N / M^2$ windows, the spatial complexity is $O((N/M^2) · M^4) = O(N · M^2)$.
>
> The combined module cost is $O(N · T + N · M^2) = O(N · (T + M^2))$. In our setting, M = 5 is fixed, thus, the complexity scales linearly with respect to $T$. A global-attention alternative over all spatial tokens would create dense interactions across $N$ tokens, causing the spatial attention term to scale quadratically as $O(N^2)$ rather than $O(N · M^2)$.
>
> We also compared NeuroMamba against a Transformer variant that replaces the Mamba temporal scan with standard multi-head self-attention. The FLOPs (Giga-operations) for varying temporal lengths (T) are summarized below:
>
> ------------------------------------------------
>  Module       |  T=5  |  T=10  |  T=20
> --------------|-------|--------|--------
>  NeuroMamba   |  42.8 |   77.7 |  147.5
>  Transformer  |  48.2 |   99.2 |  233.6
> ------------------------------------------------
>
> As T grows from 5 to 20, our FLOPs increase by a factor of 3.4 (near-linear), while the Transformer exhibits a increase of 4.8 times (super-linear). We will include this analysis in the revised paper.
>
> **Q2: Extrapolation may hallucinate under sudden motion changes**
>
> Response:
> Abrupt motion changes can indeed break pure inertia-based extrapolation and may even create phantom trajectories after occlusion. NeuroMamba handles this risk with a two-stage pipeline.
>
> In the first stage, HSR generates candidate recovery features from historical states. Its role is to restore missing information under degraded sensing, not to guarantee that every candidate is always correct.
>
> In the second stage, SCG screens candidates before they are injected into the final representation. SCG takes the latent token $z_t$  produced by HSR (Eq. 6), which carries both semantic cues and recovery-quality cues.
>
> SCG is trained with $L_{\text{guide}}$ (Eq. 12) using the GT occupancy supervision, so it learns to pass credible geometric evidence and suppress artifacts. Spatially, hallucinations in empty areas are discouraged because  $L_{\text{guide}}$ pushes the membrane response down at those locations.
>
> Temporally, the LIF dynamics (Eq. 9) act as a natural filter: strong $I_t$ opens the gate quickly, moderate $I_t$ requires multi-frame accumulation, and inconsistent extrapolations cause $I_t$ to fluctuate so leakage prevents reaching $V_{\\text{th}}$.
>
> Besides, the residual path (Eq. 11) preserves raw observations when the gate is closed, preventing incorrect candidates from overwriting the input. Table 3 shows that HSR-only improves mAP (+16.7%) but worsens mAVE (118.4→123.2), while adding SCG reduces mAVE to 104.7 (11.6% better than baseline).
>
> **Q3: $V_{\\text{th}}$ and non-stationary noise**
>
> **Response:**
>
> We address the concern from three perspectives:
>
> 1. While the threshold remains fixed, the gating behavior is adaptive because the evidence stream changes. Sensory corruption shifts the quality of HSR output, thus altering the SCG input (Eq. 6). The MLP maps this input to an evidence magnitude (Eq. 8), and the membrane integrates it with decay (Eq. 9). With the same $\\theta$, different evidence statistics create different accumulation rates and spiking outcomes.
>
> 2. Heterogeneous noise already evaluated. Our nuScenes-C benchmark (Table 1) combines depth-dependent fog, precipitation, 11 camera artifact types, and 7 LiDAR failure categories, with up to 4 concurrent effects and non-i.i.d. temporal corruptions. NeuroMamba achieves consistent improvements across all levels, demonstrating generalization under heterogeneous, non-stationary noise with fixed $V_{\\text{th}}$.
>
> 3. Threshold ablation. We provide results under medium degradation:
>
> | Threshold $\\theta$ | mAP (↑) | NDS (↑) | mAVE (↓) |
> |:---|---:|---:|---:|
> | 0.5 | 0.495 | 0.511 | 1.085 |
> | 1.0 (ours) | 0.525 | 0.527 | 1.047 |
> | 1.5 | 0.482 | 0.494 | 1.140 |
> | 1.75 | 0.502 | 0.511 | 1.095 |
> | 2.0 | 0.460 | 0.476 | 1.180 |
>
> We will include this ablation in the revised manuscript.
>
> **Q4: "Physics-guided" Terminology May Be Misleading**
>
> **Response:**
> We appreciate the reviewer's observation. The supervision signal comes from GT 3D bounding boxes projected onto the BEV plane, which is geometry-based rather than physics-based. We will replace the term "Physics-Guided" with "Occupancy-Guided" throughout the manuscript.

---

> > ### Author Rebuttal · Reviewer_opKc · 2026-04-03
> >
> > The reviewer would like to appreciate the authors' clarification on the review questions.
> > The result regarding the firing threshold is convincing enough, and the reviewer would like to thank the authors for pointing out the results that have already been shown. The rebuttal properly addresses all the questions in the review. Therefore, this led the reviewer to increase the post-rebuttal rating.

---

### Decision · Program_Chairs · 2026-04-30

**Decision:**

Accept (regular)

**Comment:**

The paper presents NeuroMemba for robust perception. Following the rebuttal, all reviewers provided positive ratings, although some noted that the work resembles an “A+B+C” combination of existing components.

After reviewing the paper, the authors’ rebuttal, and the reviewers’ comments, I agree with this assessment. While the individual components appear largely incremental, the paper offers comprehensive experimental validation, thorough model analysis, and demonstrates notable performance gains.

In light of these strengths, I recommend acceptance.